# Reconstructing Clonal Evolution—A Systematic Evaluation of Current Bioinformatics Approaches

**DOI:** 10.3390/ijerph20065128

**Published:** 2023-03-14

**Authors:** Sarah Sandmann, Silja Richter, Xiaoyi Jiang, Julian Varghese

**Affiliations:** 1Institute of Medical Informatics, University of Münster, 48149 Münster, Germany; 2Department of Computer Science, University of Münster, 48149 Münster, Germany

**Keywords:** clonal evolution, single-nucleotide variant, copy number variant, simulation

## Abstract

The accurate reconstruction of clonal evolution, including the identification of newly developing, highly aggressive subclones, is essential for the application of precision medicine in cancer treatment. Reconstruction, aiming for correct variant clustering and clonal evolution tree reconstruction, is commonly performed by tedious manual work. While there is a plethora of tools to automatically generate reconstruction, their reliability, especially reasons for unreliability, are not systematically assessed. We developed clevRsim—an approach to simulate clonal evolution data, including single-nucleotide variants as well as (overlapping) copy number variants. From this, we generated 88 data sets and performed a systematic evaluation of the tools for the reconstruction of clonal evolution. The results indicate a major negative influence of a high number of clones on both clustering and tree reconstruction. Low coverage as well as an extreme number of time points usually leads to poor clustering results. An underlying branched independent evolution hampers correct tree reconstruction. A further major decline in performance could be observed for large deletions and duplications overlapping single-nucleotide variants. In summary, to explore the full potential of reconstructing clonal evolution, improved algorithms that can properly handle the identified limitations are greatly needed.

## 1. Introduction

According to the International Agency for Research on Cancer, more than 19.2 million new cases of cancer were registered in 2020. Accounting for almost 10 million deaths, it is one of the world’s leading causes of death [1]. The survival rates are influenced by many factors, e.g., the type of cancer or the stage at which it was diagnosed. For prostate cancer, as an example, the 1-year survival rate is 97% (5-year survival rate 87%), while it is only 41% for lung cancer (5-year survival rate 17%; data for the United Kingdom [2]). Similarly, the probability of recurrence considerably differs: for childhood acute myeloid leukemia (AML), it is reported to vary between 9% and 29% [3] and, for melanoma, even between 5% and 67% (within 180 days) dependent on stage [4].

The causes of cancer are diverse. In addition to genetic predisposition, environmental factors play a major role. Exposition to physical, chemical, or biological carcinogens favor the development of mutations, which can lead to the transformation of normal cells to tumor cells [5]. For many types of cancer, the precise mutational characterization of a tumor is essential, influencing diagnosis, prognosis, and therapy. In the case of myeoldysplastic syndromes (MDS), the Revised International Prognostic Scoring System (IPSS-R)—considering, among others, the presence of cytogenetic abnormalities—is commonly used to stratify patients for high vs. low risk of AML transformation [6]. A recent study conducted by [7] provides evidence that the further subgrouping of low-risk MDS patients can be performed based on their mutational profile. For Burkitt lymphoma (BL), there is evidence that the presence of a double-hit event, that is, two variants affecting the gene *TP53*, is associated with relapse [8]. For acute lymphoblastic leukemia (ALL), therapy with a tyrosine kinase inhibitor, such as Imatinib, is assumed to be highly beneficial in the presence of an BCR:ABL translocation [9].

In addition to characterizing a tumor with respect to the presence or absence of certain variants at the time of initial diagnosis, monitoring the evolution of the mutational profile in the course of a disease is equally important. It allows for the early detection of newly developing, highly aggressive subclones that might be resistant to therapy and can potentially lead to relapse [8] (see Figure 1).

To determine a tumor’s mutational profile and its development over time, two main approaches can be distinguished: bulk and single-cell DNA sequencing (scDNA-seq). In terms of bulk sequencing, small variants are determined by, e.g., targeted or whole-exome sequencing (WES). The cancer cell fraction (CCF) estimated for each variant represents its average abundance across all the analyzed cells. The information on which variants co-occur within each cell is not directly available from the data. Instead, deconvolution has to be performed to decipher the underlying clonal populations and, subsequently, reconstruct clonal evolution. High intra-tumor heterogeneity [10,11] poses a major challenge for this approach. Unique results cannot always be retrieved. On the contrary, single-cell sequencing allows—in theory—to precisely determine the mutation profile of every analyzed cell. Technical challenges, e.g., allelic drop-out or amplification bias leading to low sensitivity [12,13], however, still hamper valid variant calling and the subsequent reconstruction of clonal evolution using this technique.

Apart from small variants, structural and copy number variants (CNVs) play a major role in clonal evolution. In addition to next-generation sequencing techniques, these variants are commonly detected using microarrays, fluorescence in situ hybridization (FISH), and/or karyotypin. While microarrays, such as SNP-arrays, only allow for a rough estimation of CCFs on the bulk level, FISH and karyotyping provide data on single-cell level. However, the resolution is considerably lower compared to scDNA-seq (100–200 interphase nucleoli for FISH, 10–25 metaphases for karyotyping). However, the valid reconstruction of clonal evolution is only possible if the data on both small and large variants are integrated and jointly evaluated [14,15].

As clonal evolution thrives to analyze cancer development over time, the samples are usually collected at several time points within the course of the disease. Solid tumors additionally hold the option for collecting multi-regional samples—within one tumor and/or at different sites (e.g., primary tumor, lymph node, and metastasis).

A plethora of approaches that perform the reconstruction of clonal evolution fully automatically is available. In a previous study, we aimed at assessing the performance of tools for the analysis of bulk sequencing data, considering two sets of well-characterized real data from the patients with MDS [16] and BL [8]. Our analysis indicated that the performance of the currently available approaches does not warrant their safe usage in research or clinical routine. The reasons for these observations, however, remained unclear. The data neither allowed us to identify those characteristics that were primarily responsible for the unreliable performance of the evaluated tools, nor to develop countermeasures [17].

To the best of our knowledge, a systematic evaluation of the tools reconstructing clonal evolution, exploring their strengths and weaknesses in relation to, e.g., the number of clones or time points, has not been performed yet. To provide a means for systematic evaluation, we developed clevRsim—a simulation approach for clonal evolution in R. clevRsim has been designed to simulate single-nucleotide variants (SNVs) in bulk sequencing data as well as (overlapping) CNVs on the basis of a user-definable clonal evolution pattern. Simulated data can then be used as an input for tools performing variant clustering and clonal evolution tree reconstruction. Considering different levels of difficulty, we simulated 88 data sets with 10 patients each. We perform a detailed systematic evaluation of nine tools for variant clustering and four tools for clonal evolution tree reconstruction that were previously identified by systematic search.

## 2. Materials and Methods

### 2.1. clevRsim

An overview of the simulation tool clevRsim is provided in Figure 2 (screenshots provided in Section A.1. clevRsim, Figure A1, Figure A2, Figure A3 and Figure A4).

Creating a simulation in clevRsim can be divided into two main steps: (1) simulating the phylogeny and (2) simulating the variants. A download option allows for exporting all the simulated data, including a CCF matrix (information on the phylogeny and CCFs at every simulated time point), variant calls (SNVs and CNVs), visualization of clonal evolution, and selected input parameters. Optionally, files can be uploaded to clevRsim again to retrieve a previous simulation. Our simulation tool is programmed in R [18]. A graphical user interface for intuitive usage also used by non-computer scientists was developed using R shiny. The software is freely available at https://github.com/sandmanns/clevRsim, accessed on 4 March 2023.

#### 2.1.1. Simulating the Phylogeny

When executing a simulation using clevRsim, the phylogeny is simulated first. As an input, the algorithm takes into account: the model of evolution, the number of time points, clones and variants (basic settings), the detection threshold, and the minimum clonal distance (advanced settings).

We consider linear, branched dependent, and branched independent evolution [8]. Additional models ‘neutral’ and ‘punctuated’ evolution, previously described by [19], can be easily simulated by defining a high number of clones for branched dependent (=neutral) or linear evolution (=punctuated).

The number of time points $n_{tp}$ defines the number of samples collected in the course of disease (default: 3 [1–50]). The number of clones $n_{c}$ corresponds to the number of clusters to be simulated (default: 5 [1–50]). Each cluster is assumed to define a distinct population of tumor cells characterized by the same mutational profile. The number of variants $n_{v}$ defines the overall number of both SNVs and CNVs to be simulated and split across the simulated clones (default: 20 [1–200]). The detection threshold $min_{th}$ defines the minimum CCF at which a clone can still be detected (default: 2 [0–50]), while the minimum clonal distance $min_{dist}$ defines the minimum CCF separating the two clones (default: 4 [0–50]).

First, the CCFs of the founding clone $c_{1}$ are simulated at random for all time points $n_{tp}$ (uniform distribution). Subsequently, the CCFs of the remaining $n_{c}-1$ clones are successively simulated considering the underlying model of evolution: For linear evolution, a new clone $c_{i}$ always develops from the latest clone simulated $c_{i-1}$. For branched dependent evolution, clone $c_{2}$ develops linearly from $c_{1}$, while $c_{3}$ develops as the first dependent branch (parent: $c_{1}$). All subsequent clones randomly develop as either linear or branched dependent. For branched independent evolution, clone $c_{2}$ develops as the first independent branch (parent: normal cells). All subsequent clones randomly develop as either linear or branched independent.

For every new clone, its CCFs are simulated under the following constraints: A child clone cannot exceed its parent at any time. The sum over all dependent child clones cannot exceed their joined parent. The sum over all independent clones cannot exceed 100%. An eradicated clone cannot re-appear, i.e., once a clone’s CCF has fallen to zero, it remains zero for all subsequent time points. The CCF of each clone has to be $\geq min_{th}$ for at least one time point. The Δ CCF for two clones has to be $\geq min_{dist}$ for at least one time point. If one of the constraints is violated, a new clone is simulated at random. If no solution is found after 100 attempts, a new simulation, including a new founding clone, is executed (maximum 700 attempts). Finally, if a phylogeny was successfully generated, one variant is assigned to every clone. The remaining $n_{v}-n_{c}$ variants are randomly split.

The results of the algorithm simulating the phylogeny are visualized using the R/Bioconductor package clevRvis [20]. Additionally, a CCF table is displayed, providing an option for fine-tuning the simulation on a clonal level. All major characteristics of the clonal evolution (parental relations, number of variants per clone, and CCFs) can be changed, and the clones and time points can be added as well as deleted.

#### 2.1.2. Simulating Variants

As a second step when executing a simulation with clevRsim, the genetic variants are simulated under the infinite-allele assumption [21]. The simulation of CNVs is executed first. As an input, the algorithm takes into account the previously simulated phylogeny including the CCFs of every clone at every time point and the number of variants assigned to each clone. When simulating SNVs, the mean coverage, purity, and minimum clonal distance (advanced settings), as well as the simulated CNVs, are additionally considered.

For each CNV, a genomic location is generated at random (chromosomes 1–22, positions according to GRCh37, random CNV length, but ≤10,000 bp). clevRsim supports the simulation of deletions, duplications, and loss of heterozygosity (LOH). Every selected type of CNV is simulated at least once. Subsequently, the types of the remaining CNVs are determined at random. The CNVs are randomly assigned to the simulated clones.

Subsequently, the SNVs are generated. In the absence of CNVs, SNVs are simulated completely at random (chromosomes 1–22, positions according to GRCh37). In the presence of CNVs, all the SNVs are initially simulated as “non-overlapping”. If the genomic locations of the simulated SNVs overlap the CNVs by chance, the simulation of the corresponding SNVs is repeated. The reads are simulated on the basis of a log–normal distribution (μ=log(user-defined mean coverage), σ=log(0.7); the standard deviation σ was determined empirically to best approximate the coverage distribution, evaluating the exemplary targeted and WES data [8,16]). The variant allele frequency (VAF) is determined as VAF=CCF/2. Random variation is added (N(0,1)) to account for imperfect sequencing.

The results of the simulated variants are tabulated. For every call, information on the basic and advanced characteristics—also including the number of simulated reads and VAF for SNVs—is provided. Based on these results, the overlap of CNVs and SNVs can be configured (Fine-tuning: CNVs). This includes the number of overlapping SNVs, the affected clone as well as the scenario of overlap. We distinguish four scenarios: CNV first, SNV first affected, SNV first un-affected, and parallel [14] (the influence of the different scenarios on the genotype is summarized in Section A.1. clevRsim, Table A1). According to the formula provided in [14], the VAF is adjusted with respect to the scenario and CNV type. For example, in the case of a deletion present in the scenario ‘SNV first un-affected’, we expect to observe the cells characterized by SNV (genotype AB) and SNV+CNV (genotype B). Assuming that the CCF of clone $c_{SNV}$ is 100% and the CCF of clone $c_{SNV+CNV}$ is 50%, the adjusted VAF (that is assumed to be observed in a sequencing experiment) is 67%, not 50%. The number of simulated reads is adjusted accordingly. The overlapping SNVs are randomly positioned within the corresponding CNVs. A clone is assigned at random, restricted only by the selected scenario of overlap and the number of variants per clone.

### 2.2. Simulated Data Sets

Altogether, 88 data sets were simulated using clevRsim. An overview of the data sets and their main characteristics is provided in Table 1. To account for random variation in the simulated data and their influence on the tools’ performance, we simulated ten patients per data set leading to a total of 880 simulated patients.

The simulated data sets can be split into 3 groups. The first group—from sim01 to sim40—serves as a basic evaluation of the tools performing variant clustering. The data are characterized by a varying number of time points (from 1 to 10), number of clones (from 1 to 10), number of SNVs (from 5 to 50 in steps of 5), and mean coverage (from 10x to 2000x). For all sets, we chose ‘linear’ as the model of evolution. CNVs were not simulated.

The second group—from sim41 to sim52—serves an advanced evaluation of the tools performing variant clustering, also considering the present CNVs. As we could not identify any algorithm that was capable of clustering the SNVs and non-overlapping CNVs, only overlapping CNVs were simulated. Every data set contains 20 SNVs and 6 overlapping CNVs: 3 CNVs are overlapping 1 SNV each, 2 further CNVs are overlapping 2 SNVs each, and 1 CNV is overlapping 3 SNVs. Thereby, 10 out of 20 simulated SNVs per data set are affected by the overlapping CNVs. For every data set, we chose a different scenario of overlap (CNV first, SNV first affected, SNV first un-affected, and parallel).

The third group—from sim53 to sim88—serves, together with simulated data sets from sim01 to sim20, an evaluation of the tools performing clonal evolution tree reconstruction. Commonly, the tools evaluate the information on clones and their CCFs at every time point. Therefore, we evaluate a varying number of time points (from 1 to 10) and number of clones (from 3 to 10), considering all three models of evolution: linear, branched dependent, and branched independent. Of note, the branched dependent and branched independent evolution cannot be simulated in the presence of only 1 or 2 clones as a branch requires at least 3 clones.

### 2.3. Tools for Variant Clustering

We base our systematic evaluation of tools for the reconstruction of clonal evolution on a previously performed evaluation using two sets of real data [17]. In our previous study, the systematic search identified 40 algorithms performing variant clustering. However, 29 out of the 40 tools had to be excluded from the detailed evaluation as they had a different scope, were unavailable, outdated, defective, or had missing documentation, which caused interpretation of the results to be impossible.

Aiming at identifying the reasons for the previously observed poor performance of the tools on real data, we reconsidered the remaining 11 tools cloneHD [22], clonosGP [23], DeCiFer [24], PyClone [25], PyClone-VI [26], QuantumClone [27], sciClone [28], Canopy [29], Cloe [30], LICHeE [31], and SPRUCE [32]. However, three tools had to be excluded from the detailed evaluation: cloneHD (missing clustering information in the output), LICHeE, and SPRUCE (both: high level of missing data). Accounting for recently published approaches performing variant clustering, the tool PICTograph [33] was additionally considered.

Altogether, a detailed evaluation of 9 tools performing variant clustering at different scenarios was executed (see Section A.2 for the tools for variant clustering and detailed information on all tools).

### 2.4. Tools for Clonal Evolution Tree Reconstruction

Similar to the systematic evaluation of tools for variant clustering, we reconsidered the previously evaluated tools for clonal evolution tree reconstruction. In our study, evaluating real data, we identified 30 algorithms performing this task. However, the majority of the 22 tools had to be excluded due to having a different scopes, being unavailable, outdated, defective, or lacking documentation. Furthermore, 4 algorithms (Canopy, Cloe, LICHeE, and SPRUCE) were excluded from detailed evaluation as they must perform tree reconstruction on their own (partly erroneous) variant clustering results. The remaining tools taken into account were ClonEvol [34], ClonalTREE [35], SCHISM [36], and TrAP [37].

Two additional recently published tools were identified: ClonalTREE2 [38] and SubMARine [39]. However, both tools were not considered for detailed evaluation (ClonalTREE2: additionally performing variant clustering; SubMARine: not reporting explicit clonal evolution trees for a majority of samples) (see Section A.3 for tools for clonal evolution tree reconstruction and detailed information on all tools).

### 2.5. Statistical Analysis

The statistical analysis was performed using R 4.1.2 [18]. To evaluate the tools’ performance on variant clustering, we determined the variation of information (VI), using the R package ‘clevr’ [40]. The metric allows for calculating the entropy-based distance between two clusterings, with lower values corresponding to greater similarity.

To evaluate the tools’ performance on clonal evolution tree reconstruction, we determined the discrete spectral distance (DSD), using the R package ‘NetworkDistance’ [41]. We defined the clonal evolution trees as directed graphs using the R package ‘igraph’ [42]. Subsequently, the graphs were transformed to adjacency matrices, for which the DSD was calculated as a similarity measure.

## 3. Results

### 3.1. Reliability of clevRsim

The simulation tool clevRsim was used to generate 88 data sets (880 patients) on clonal evolution. These sets form the basis for a systematic evaluation of tools reconstructing clonal evolution. Both variants and phylogeny are simulated at random—only restricted by user-defined input parameters and basic constraints of clonal evolution. To verify that our simulated data warrants an unbiased evaluation of the tools, the reliability of clevRsim is investigated.

With respect to variants, it can be observed that all SNVs are randomly split over the simulated clones. This observation holds for varying numbers of time points, clones, variants, and coverage (see Section B.1. The reliability of clevRsim, Figure A5). Exemplarily considering a varying number of clones, no biasing influence of the underlying model of evolution can be observed (Figure A6).

Considering phylogeny, a random number of branches is simulated for both the branched dependent and independent evolution (see Section B.1. The reliability of clevRsim, Figure A7). No significant difference can be observed comparing the two models of evolution. Of note, the average number of simulated branches is usually lower than the maximum number of possible branches. However, this behavior of clevRsim was intended as branched evolutionary patterns were observed to commonly consist of both branched and linear development (e.g., [8,16]).

Regarding the development of CCF over time, an influence of the clone considered as well as the model of evolution can be observed, indicating the correct functioning of clevRsim (see Section B.1. The reliability of clevRsim, Figure A8). The first clone (stemline) is commonly characterized by the highest CCFs. For linear clonal evolution, subsequent clones follow with gradually decreasing CCFs as a child clone cannot exceed its parent. For both branched dependent and independent evolution, however, the variation in the CCFs of the subsequent clones can be observed, indicating the correct independent development of clones belonging to different branches.

### 3.2. Variant Clustering in the Absence of CNVs

To evaluate the basic performance of tools for variant clustering, we simulated 40 data sets (400 patients) only containing SNVs. We consider three time points, five clones, 20 variants, and a coverage of 300x as our baseline configuration. From the simulation, we analyze the influence of each of the four parameters on the variant clustering. The results are summarized in Figure 3 (see Section B.2. For the variant clustering in the absence of CNVs, see Table A2 for the exact numbers; information on the available clustering data is provided in Figure A9; the reported numbers of clusters for each tool and data set are provided in Figure A10).

When manually performing variant clustering, the results commonly improve as the number of time points increases. The more information available on the variants and their development of CCFs over time, the more clearly the clusters emerge. In Figure 3a, it can be observed that the number of available time points has only a small but diverse effect on the tools’ performance. clonosGP, QuantumClone, Canopy, and Cloe show the expected behavior of the decreasing VI as the number of time points increases. The remaining tools, on the contrary, show a minor (PyClone, PyClone-VI, and PICTograph) up to major (sciClone and DeCiFer) increase in dissimilarity compared to the ground truth. However, most tools share poor performance in the presence of only one time point.

Regarding an increasing number of clones, we expect the performance of the tools to decrease. The more clones are present, the higher the chance of clones showing similar evolution with only minor differences in CCF. All the considered tools show the expected behavior (see Figure 3b). It is interesting to observe that, even in the presence of only one clone, some tools such as sciClone do not safely succeed in performing perfect clustering but instead overestimate the number of clones (average number of clones 2; $VI_{sciClone}=0.50$).

For the number of variants, it can be observed that their influence on most tools is mainly insignificant (see Figure 3c). Just in the case of clonosGP, an improved performance in the presence of more variants can be observed, while PICTograph shows poorer performance.

With respect to coverage, we expect to observe improved performance of the tools as coverage increases. Low coverage increases the variation in the estimated CCFs, characterizing the SNVs to cluster. As these estimated CCFs may thus show considerable differences from the true CCFs, valid clustering is challenging. As expected, all the tools show a decrease in VI as the coverage increases (see Figure 3d). The only exception from this development is PICTograph, showing the best results between 20x and 100x. Of note, the data indicate a general decline in performance at a very high coverage of 2000x.

To analyze whether the observed differences in the performance do not just result from random variation, we considered the robustness of the results at the exemplary case of the varying number of time points. The evaluation of the quartiles $Q_{1}$ and $Q_{3}$ indicates that the true differences in the performance exist and that the simulated number of n=10 patients per configuration is apt to observe and assess these differences (see Section B.2. For the variant clustering in the absence of CNVs, see Figure A11.

### 3.3. Variant Clustering in the Presence of CNVs

To evaluate the performance of tools for variant clustering in an advanced, more challenging setting, we simulated 12 data sets (120 patients) containing 20 SNVs and and 6 overlapping CNVs each. Figure 4 sums up the results, considering deletions, duplications, and LOH in comparison to the baseline configuration just containing 20 SNVs (see Section B.3. For variant clustering in the presence of CNVs, see Table A2 for the exact numbers; information on the available clustering data is provided in Figure A12; the reported numbers of clusters for each tool and data set are provided in Figure A13).

We aimed at simulating the data as close as possible to real data. We therefore assumed that the SNVs and CNVs were detected by performing different experiments, e.g., WES and SNP arrays. Consequently, we only know about the presence of (overlapping) variants, but not necessarily about the underlying scenario of the overlap. If a deletion and an SNV are detected in one clone, it is unclear whether the underlying genotype is B (CNV first or SNV first un-affected), A (SNV first affected), or AB and A (parallel; compare Appendix A, Table A1). For all tools, therefore, we just defined the copy number as ‘1’ instead of ‘2’ for the SNVs overlapping deletions. Duplications and LOH were handled accordingly.

It can be observed that, in the presence of deletions and duplications, the performance of all tools is considerably worse compared to our baseline configuration. The differences between the scenarios of overlap are usually small. However, the data indicate the lowest performance for scenarios that are SNV first (affected and un-affected) for most tools.

Considering the influence of LOH, different results can be observed. In general, the tools’ performances are much better and even comparable to baseline. However, as the copy number is 2 and the genotype is AB for two out of four scenarios, these results are mainly expected.

### 3.4. Clonal Evolution Tree Reconstruction

The performance of tools for clonal evolution tree reconstruction is evaluated by 56 data sets (560 patients), considering varying numbers of time points and clones. For each, we simulated the underlying linear, branched dependent, and branched independent clonal evolution. The results are summed up in Figure 5 (see Section B.4. For the clonal evolution tree reconstruction, see Table A3 for the exact numbers; information on the available clustering data is provided in Figure A14; the reported numbers of clusters for each tool and data set are provided in Figure A15).

Regarding the influence of the available time points, a slight tendency toward better performance can be observed as the number of time points increases (Figure 5a,c,e). For ClonEvol and TrAP, the data indicate a decline in the performance as more challenging clonal evolution patterns are simulated (linear > branched dependent > branched independent). In the context of branched independent evolution, ClonEvol only succeeds in reconstructing clonal evolution trees for 2 out of 10 settings (2 and 3 time points; analysis of 1 time point generally not supported), while SCHISM shows a particularly poor performance, independent of the number of time points. ClonalTREE is the only tool showing comparable performance across all underlying models of clonal evolution.

With respect to a varying number of clones, the behavior of the tools for tree reconstruction is comparable to the tools for variant clustering (Figure 5b,d,f). An increasing number of clones leads to an increasing DSD and, thus, to the lower performances of the tools. Comparing linear vs. branched dependent evolution, the performances of the tools are usually similar. Considerable differences can, however, be observed for branched independent evolution, while SCHISM shows especially poor performance in the presence of ≥6 clones, ClonEvol and TrAP do not succeed in generating any trees for seven or more clones. ClonalTREE is the only tool showing a relatively stable performance independent of the underlying model of evolution.

## 4. Discussion

In this work, we performed a systematic evaluation of tools for variant clustering and clonal evolution tree reconstruction. To our knowledge, a comparable thorough analysis has not been performed yet. With the help of simulated data sets generated with our novel approach clevRsim, we analyzed the influence of a varying number of time points, clones, SNVs, coverage, CNVs, and the underlying model of evolution.

Our results indicate that a high number of clones poses a major challenge for all variant clustering tools. This observation is in line with our previous results on real dat: the correct clustering could not be automatically determined for any patient characterized by >3 clones [17]. Regarding clinical practice, this observation implies a considerable challenge. Several studies reported on the high level of intra-tumor heterogeneity to be observed in many cancers, e.g., [10]. However, heterogeneity is expected to hamper the valid automatic variant clustering, leading to an underestimated number of clusters.

For a varying number of time points, diverse tool-dependent effects could be observed. A majority of tools showed poor performance in the presence of only one time point. With respect to clinical practice, this observation may be similarly challenging. At the beginning of a disease, only limited amount of data are available. However, in order to use clonal evolution analysis to support treatment decisions, valid results—even in the presence of only one time point—are needed.

A clear negative influence on the tools’ performance could be observed for low coverage. This challenge may be overcome by increasingly performing deep targeted sequencing or—in case information on commonly mutated hotspot genes is lacking—high-coverage WES. However, it was also observed that a very high coverage of 2000x again leads to performance degradation. The detailed analysis of the results showed that variants of one cluster are often split among multiple clusters for all tools. It appears likely that the reason for this observation is the “over-interpretation” of small differences in the estimated CCFs. However, unexpectedly, we could also identify some clusters, partly showing differences of >20% CCF, being merged. This observation remains unclear as higher coverage is expected to lead to higher confidence in the estimated CCFs.

A particularly negative impact on variant clustering could be observed for deletions or duplications overlapping SNVs. For all tools, dissimilarity between the proposed and the true clustering was considerably increased. Several issues underlie this observation: First, a majority of the tools just allow for defining the underlying copy number for every SNV and not the fraction of cells characterized by a certain copy number. Second, the scenario of overlap is not considered despite having a major influence on the genotype and requiring an adjusted calculation of the CCF (formula provided in [14]). Third, none of the considered tools are capable of jointly clustering CNVs and SNVs in the case of lacking overlap.

Reconstructing clonal evolution trees on the basis of correctly clustered variants is not free from flaws either. The presence of a high number of clones as well as branched independent evolution increased the dissimilarity between the correct and the reported trees. When introducing their tool, SCHISM, [36] observed similar results: in the presence of many clones, SCHISM only succeeded in reporting the correct tree under ideal conditions—many time points, high coverage, and high purity of the samples. However, it should be mentioned that [36] just determined whether the correct tree was upon all trees reported by SCHISM. For 55% of all the simulated patients in our study, the tool reported an ambiguous consensus tree.

Furthermore, all the results we observed on tree reconstruction in our study were generated in an optimized setting. Commonly, the output of variant clustering tools is provided as input, which—as our results showed—does not necessarily match the correct clustering. Thus, moving from a list of variants to properly reconstructed clonal evolution, errors are expected to multiply.

To provide a means for developing new optimized algorithms, our simulation tool clevRsim is expected to provide a valuable input. Randomly simulating phylogeny and variants, our approach is able to generate an unlimited number of valid clonal evolution patterns. As future work, we plan to extend clevRsim considering a user-defined input on clinical parameters, which is expected to modulate the birth and death rates of simulated cell populations. For example, a therapy being applied is likely leading to a general decrease in CCFs and an eradication of some cell populations, while relapse is probably preceded by several new (branching) clones. Furthermore, the simulation of driver vs. passenger mutations and their effect on CCF development will be explored. As a result, the future updated version of clevRsim will allow for generating more specific clonal evolution patterns automatically—without the need for manual fine-tuning that exists at present.

It can be discussed why we focused our evaluation of bioinformatics approaches for reconstructing clonal evolution on the—compared to scDNA-seq—relatively old technique of bulk sequencing. Despite bearing the potential to revolutionize the field of clonal evolution, scDNA-seq is currently still characterized by major technical and practical challenges. Allelic drop-outs and amplification bias cause low sensitivity for detecting variants. Sequencing costs are high, which hampers its use in large study cohorts or even clinical routine. Additionally, missing suitable material prohibits the retrospective analysis of samples. As a consequence, we expect that the reconstruction of clonal evolution on the basis of bulk sequencing data will continue to play a major role within the next decade.

## 5. Conclusions

With respect to optimizing cancer treatment by the detailed study of the underlying clonal evolution, our study shows that the current algorithms for fully automatic reconstruction are still characterized by major flaws. While the coverage of the sequencing experiment can be actively influenced, the tools’ inability to perform correct clustering and tree reconstruction in the presence of many clones and few time points is likely to lead to false results. The same is true for additionally present, possibly overlapping CNVs. There is a pressing need for better algorithms, automatically determining the underlying scenario of overlap as well as jointly clustering CNVs and SNVs with and without overlap. Only when these current limitations are overcome and clonal evolution does no longer have to be reconstructed by tedious manual work, its full potential may be explored.

## Figures and Tables

**Figure 1 ijerph-20-05128-f001:**
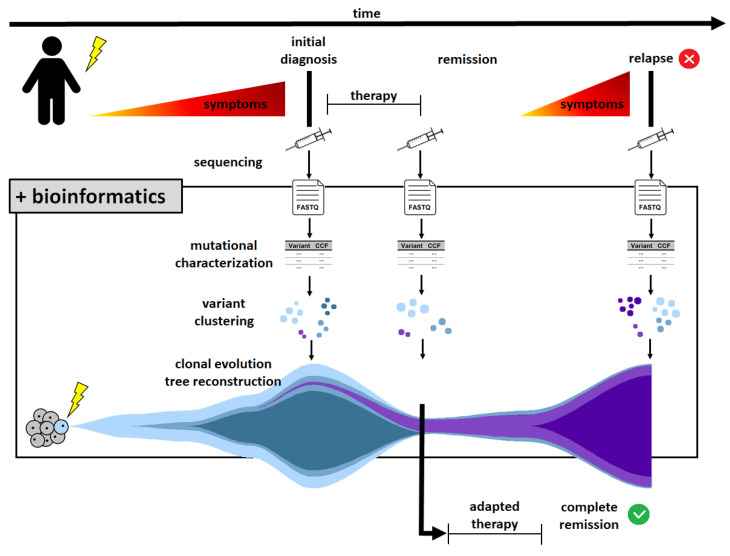
The role of bioinformatics in precision oncology. When performing sequencing experiments in the course of disease, bioinformatics analyses allow for performing mutational characterization of a tumor. Subsequently, data can be used to reconstruct clonal evolution, including variant clustering and clonal evolution tree reconstruction. Thereby, aggressive subclones (purple), which are resistant to therapy and may lead to relapse, can be identified early and therapy adapted accordingly.

**Figure 2 ijerph-20-05128-f002:**
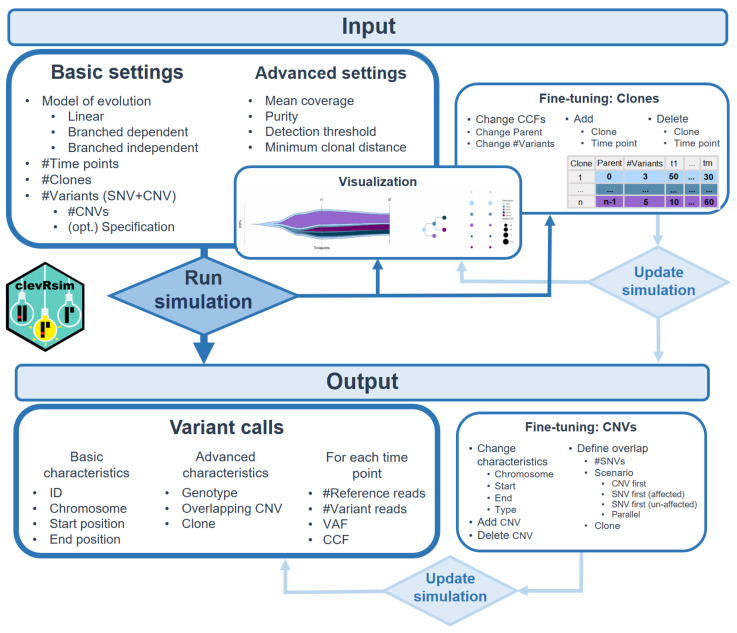
Overview of the simulation tool clevRsim. A graphical user interface is provided to generated customizable clonal evolution patterns. Basic information, including clones, their parents, and the cancer cell fraction at every time point, and a visualization of clonal evolution are provided. Additionally, variant calls—supporting SNVs as well as (overlapping) CNVs—are simulated. Output generated using clevRsim may serve as direct input for tools performing variant clustering and clonal evolution tree reconstruction.

**Figure 3 ijerph-20-05128-f003:**
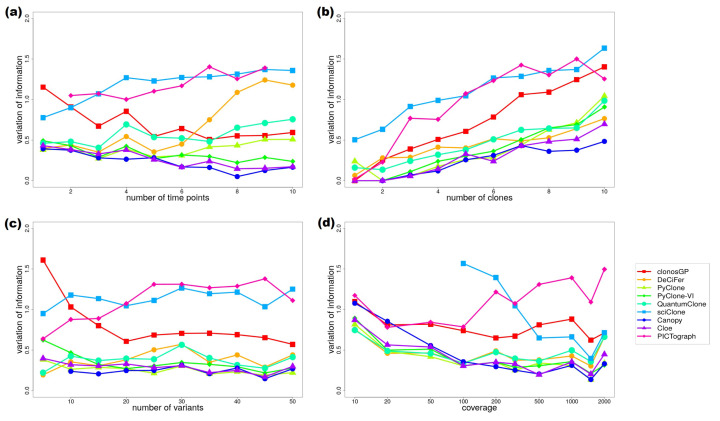
Variation of information comparing variant clustering performed using clonosGP, DeCiFer, PyClone, PyClone-VI, QuantumClone, sciClone, Canopy, Cloe, and PICTograph in the absence of CNVs to simulated truth. Three time points, five clones, 20 variants, and 300x coverage define the baseline configuration. We evaluate the influence of varying (**a**) number of time points; (**b**) number of clones; (**c**) number of variants; and (**d**) coverage. Of note, a missing dot indicates missing clustering results for all 10 patients per simulated data set.

**Figure 4 ijerph-20-05128-f004:**
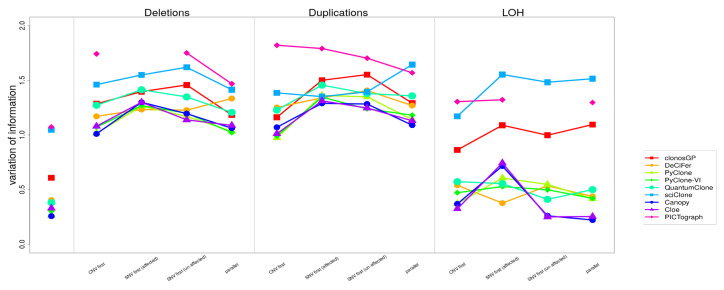
Variation of information comparing variant clustering performed using clonosGP, DeCiFer, PyClone, PyClone-VI, QuantumClone, sciClone, Canopy, Cloe, and PICTograph in the presence of CNVs to simulated truth. We evaluate the influence of deletions, duplications, and LOH, considering four scenarios of overlap: CNV first, SNV first (affected), SNV first (un-affected), and parallel. For comparison, the performance of each tool at the baseline configuration is included. Of note, a missing dot indicates missing clustering results for all 10 patients per simulated data set.

**Figure 5 ijerph-20-05128-f005:**
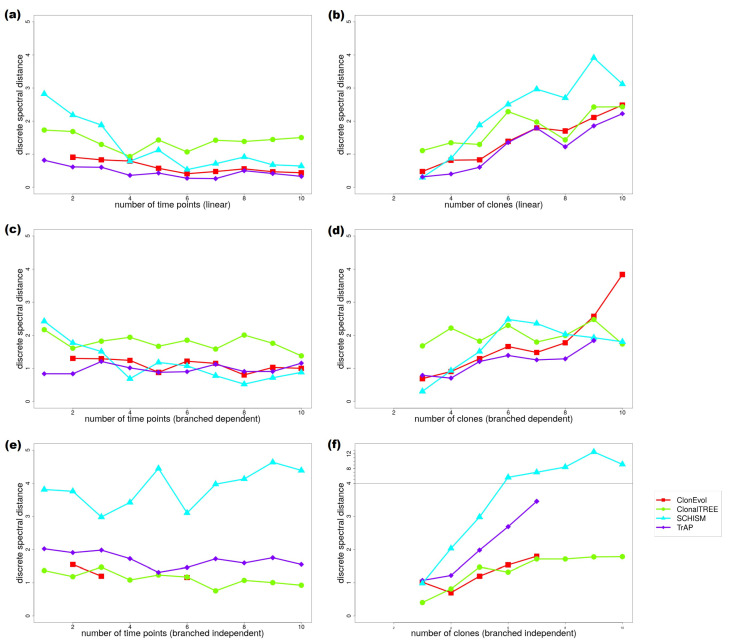
Discrete spectral distance comparing the trees reconstructed usinh ClonEvol, ClonalTREE, SCHISM, and TrAP to simulated truth. If tools report more than one tree, the average distance over all reported trees is calculated. We evaluate the influence of varying (**a**) number of time points at linear evolution; (**b**) number of clones at linear evolution; (**c**) number of time points at branched dependent evolution; (**d**) number of clones at branched dependent evolution; (**e**) number of time points at branched independent evolution; and (**f**) number of clones at branched independent evolution. Of note, branched dependent and independent evolution can only be considered in the presence of ≥3 clones. A missing dot indicates missing clustering results for all 10 patients per simulated data set.

**Table 1 ijerph-20-05128-t001:** Data sets simulated using clevRsim. Altogether, 88 data sets with 10 patients each were simulated. Simulated sets were used to evaluate performance of variant clustering tools in a basic (from sim01 to sim40) and an advanced setting (from sim41 to sim 52). Additionally, the performance of tools for clonal evolution tree reconstruction was evaluated (from sim01 to sim20 and from sim53 to sim88).

Data Set	Model of	#Time Points	#Clones	#SNVs	Mean Coverage	#CNVs
	Clonal Evolution					
sim01–sim10	linear	1–10	5	20	300x	0
sim11–sim20	linear	3	1–10	20	300x	0
sim21–sim30	linear	3	5	5–50 (step 5)	300x	0
sim31–sim40	linear	3	5	20	10x, 20x, 50x, 100x, 200x,	0
					300x, 500x, 1000x, 1500x, 2000x	
sim41–sim44	linear	3	5	20	300x	6 (del)
sim45–sim48	linear	3	5	20	300x	6 (dup)
sim49–sim52	linear	3	5	20	300x	6 (LOH)
sim53–sim62	branched dependent	1–10	5	20	300x	0
sim63–sim70	branched dependent	3	3–10	20	300x	0
sim71–sim80	branched independent	1–10	5	20	300x	0
sim81–sim88	branched independent	3	3–10	20	300x	0

## Data Availability

All data necessary to reproduce the reported results are available with the article. The approach clevRsim, used to simulate the data sets analyzed, is freely available at https://github.com/sandmanns/clevRsim, accessed on 4 March 2023.

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
