# Peer review of "Reconstructing Clonal Evolution—A Systematic Evaluation of Current Bioinformatics Approaches"

_ijerph, 2023, doi:10.3390/ijerph20065128_

Round 1

Reviewer 1 Report

Dear Sandmann and colleagues,

Your manuscript, which I have the pleasure of reviewing, is a systematic assessment of various tools for reconstructing clonal evolution in cancer. To accomplish this, the authors first simulate genomic mutations (SNV and CNA) within a clonal evolution framework using clevRsim, a simulation tool created by the authors. After that, the authors evaluate various tools for both tree reconstructions and clustering variants. In terms of the statistical method used to evaluate the various tools, the manuscript is reliable, useful, and solid. 

Even so, the authors' conclusions are expected, and in addition, as the authors point out, we are well aware of the pressing need for better algorithms.  The manuscript is deserving of consideration, but there are some suggestions I want to mention. Please see my comment below.

1) Despite an accurate introduction, I believe it is lacking in focus. The introduction ought to place more emphasis on the clonal evolution of cancer than on its epidemiology. Please revise it to reflect that.

2) Phylogenetic reconstructions from bulk tumour and single cells are currently the two main types, in addition, a third type based on multiregional sequencing of bulk/metastasis tumour samples (spatial genomics). The authors appear to discuss the inference of clonal evolution using whole bulk tumour samples, so this should be clarified. It is necessary for the authors to state that they are evaluating the clustering method and phylogenetic reconstructions on large bulk samples.

3) To simulate clonal evolution, the author created their own tool. The tool named  clevRsim  is written in R and designed to simulate single-nucleotide variants (SNVs) and (overlapping) CNVs.

Despite the author's description of clevRsim's overview and their provision of a link to the github page, in my opinion they have neglected to provide the fundamental statistics, the algorithms behind, required to evaluate the clevRsim tool.

The evaluation of the clustering and tree reconstruction tools depends on the mutations fed by clevRsim and it  is essential to demonstrate the robustness of clevRsim.

Instead of just giving a general description, the authors should go into great detail about clevRsim. I am concentrating on this point because as I read the manuscript, a few straightforward questions start to cross my mind. Consider this:

How do SNV and CNV simulations take place? Are SNVs and CNVs drawn from a limited or unlimited population of cells? Or based on a range of values? And in that scenario, what type of distribution? Alternatively, does clevRsim simply generate SNVs and CNVs without sampling? Which algorithm is then employ? posterior probabilities?

Do the authors operate under the infinite-allele premise when using clevRsim?

How do the authors distinguish between mutations in drivers and passengers?  

Do the authors use distinct types of populations and random variables to represent the birth and death rates of cell populations, or this is assumed to be neglected? 

I believe that a better description of the tool is required in order to avoid the questions mentioned above and other possible reader inquiries.

4) Model of clonal evolution

The authors model clonal evolution as linear and branching (branched dependent + branched independent), simulating SNV and CNV under each model. What process produces the phylogeny? Under a parsimonious framework? A maximum likelihood approach? Please explain

5) The meaning of the number of clones (x-axis) in Figure 3b and Figure 4 (b, d, and f) is beyond my comprehension.  According to my understanding, 2 means that there are 2 clonal entities in the cancer population, followed by 4 and so on. But how dissimilar are those clonal populations at each point in terms of mutational load and population size? What does the author believe? Please clarify; I could be misinterpreting and missing something. My observation may be more relevant to branching evolution, where I anticipate different clonal population sizes.

Thank you

Author Response

1) Despite an accurate introduction, I believe it is lacking in focus. The introduction ought to place more emphasis on the clonal evolution of cancer than on its epidemiology. Please revise it to reflect that.

Changes: We revised and extended our Introduction, now focusing on the clonal evolution of cancer.

2) Phylogenetic reconstructions from bulk tumour and single cells are currently the two main types, in addition, a third type based on multiregional sequencing of bulk/metastasis tumour samples (spatial genomics). The authors appear to discuss the inference of clonal evolution using whole bulk tumour samples, so this should be clarified. It is necessary for the authors to state that they are evaluating the clustering method and phylogenetic reconstructions on large bulk samples.

Changes: We extended our Introduction, comparing different approaches for determining a tumor's mutational profile (bulk vs sc). Additionally, we consider the case of samples collected at several time points and different tumoral regions. We added information that both our previous and our current study focus on the analysis of bulk sequencing data. Additionally, we added a paragraph to our Discussion, considering the focus of our work and the expected development of sc sequencing in the field of reconstructing clonal evolution.

3) To simulate clonal evolution, the author created their own tool. The tool named “clevRsim” is written in R and designed to simulate single-nucleotide variants (SNVs) and (overlapping) CNVs. Despite the author's description of clevRsim's overview and their provision of a link to the github page, in my opinion they have neglected to provide the fundamental statistics, the algorithms behind, required to evaluate the clevRsim tool. The evaluation of the clustering and tree reconstruction tools depends on the mutations fed by clevRsim and it is essential to demonstrate the robustness of clevRsim. Instead of just giving a general description, the authors should go into great detail about clevRsim. I am concentrating on this point because as I read the manuscript, a few straightforward questions start to cross my mind. Consider this: How do SNV and CNV simulations take place? Are SNVs and CNVs drawn from a limited or unlimited population of cells? Or based on a range of values? And in that scenario, what type of distribution? Alternatively, does clevRsim simply generate SNVs and CNVs without sampling? Which algorithm is then employ? posterior probabilities? Do the authors operate under the infinite-allele premise when using clevRsim? How do the authors distinguish between mutations in drivers and passengers? Do the authors use distinct types of populations and random variables to represent the birth and death rates of cell populations, or this is assumed to be neglected? I believe that a better description of the tool is required in order to avoid the questions mentioned above and other possible reader inquiries.

Changes: We re-wrote and extended the Methods section “clevRsim”, now providing an extensive description of the underlying algorithms. Subsections “Simulating the phylogeny” and “Simulating variants” were added. clevRsim performs random simulation of the phylogeny, including CCFs for every clone and time point. Subsequently, SNVs and CNVs are simulated under the infinite-allele premise. Variants are randomly assigned to a clone, coverage is simulated on the basis of previously analyzed real data sets (da Silva-Coelho et al. 2017, Reuter et al. 2021). A paragraph on extending clevRsim was added to our Discussion, considering driver vs passenger mutations and modulating birth and death rates of cell population in relation to user-defined clinical information. Furthermore, we added a paragraph to the Results section “Variant clustering in the absence of CNVs”, considering the robustness of clevRsim in the exemplary case of varying number of time points (details provided in Appendix B.2., Figure S11).

We would like to thank the reviewer for his very valuable input on working out the details of the algorithms implemented in clevRsim! Our approach simulates the phylogeny first and – subsequently – adds variants to the scaffold of clonal evolution. Clones are defined at random, based on an initially unlimited population of cells. Further clones are successively added accounting for the user-defined model of evolution (CCFs drawn from uniform distribution). As a consequence, the SNVs and CNVs are not directly drawn from a population of cells. Their characteristics are specified by the clone they were randomly assigned to.

4) Model of clonal evolution: The authors model clonal evolution as linear and branching (branched dependent + branched independent), simulating SNV and CNV under each model. What process produces the phylogeny? Under a parsimonious framework? A maximum likelihood approach? Please explain

Changes: We re-wrote and extended the Methods section “clevRsim”. A new subsection “Simulating the phylogeny” was added. Detailed information on how clevRsim simulates the phylogeny is now provided. In short, a phylogeny is generated at random, considering only the user-defined input and basic constraints of clonal evolution (e.g. a child-clone cannot exceed its parent at any time). We do not judge whether, at a certain point of evolution, development of a branching clone is more likely than another linear clone. However, this is a very interesting point regarding future work. We added a paragraph on extending clevRsim to our Discussion (e.g. accounting for clinical information such as a therapy being applied or relapse taking place, which will influence the development of phylogeny and CCFs).

5) The meaning of the number of clones (x-axis) in Figure 3b and Figure 4 (b, d, and f) is beyond my comprehension. According to my understanding, 2 means that there are 2 clonal entities in the cancer population, followed by 4 and so on. But how dissimilar are those clonal populations at each point in terms of mutational load and population size? What does the author believe? Please clarify; I could be misinterpreting and missing something. My observation may be more relevant to branching evolution, where I anticipate different clonal population sizes.

Changes: To evaluate the dissimilarity of the simulated data, section “Reliability of clevRsim” was added to the Results. The reliability of data generated with clevRsim is evaluated considering the simulation of variants (number of SNVs per clone for varying numbers of time points, clones, variants and coverage as well as for branched dependent and independent evolution at a varying number of clones), the simulation of phylogeny (number of branches for branched dependent vs independent evolution) and the development of CCF over time (linear vs branched dependent vs branched independent evolution at a varying number of clones). Detailed results are available as Appendix B.1. Reliability of clevRsim. Additionally, we revised and extended Methods section “clevRsim” to clarify the meaning of the “number of clones”.

We are very sorry for the confusion caused by the formulation “number of clones”. This parameter indicates the true number of clusters being simulated with clevRsim. Thus, it corresponds to the different number of clones that can be identified in a clonal evolution, each characterized by a specific mutation pattern. We hope that our revised Methods section now gives a clearer definition of the term.

Reviewer 2 Report

In the manuscript entitled "Reconstructing Clonal Evolution – a Systematic Evaluation of Current Bioinformatics Approaches" by Sarah Sandmann and colleagues, authors describe an R tool clevRsim for simulating data that can be used to model clonal evolution patterns by the tools performing automated variant clustering and clonal evolution modeling. Authors generated 88 simulated datasets and used the simulated data to perform systematic evaluation of the available tools. Overall, manuscript is well written and of great importance for the field of cancer evolution. Furthermore, the tool authors developed holds a great promise as an aid in improving existing and developing new tools dealing with clonal evolution modeling.

 Minor comments:

 1) Rearrangements between the genes should be written with "::", instead of hyphen, e.g., BCR-ABL1 should be written as BCR::ABL1.

 2) What was the reason for automated clustering tools to underperform with very high depth sequencing data? It might be of interest to comment on this in the discussion, since many studies dealing with clonal evolution modeling confirm presence of somatic mutations discovered using WES or WGS by ultra-high depth targeted sequencing.

Author Response

1) Rearrangements between the genes should be written with "::", instead of hyphen, e.g., BCR-ABL1 should be written as BCR::ABL1.

Changes: Changed to “::”.

2) What was the reason for automated clustering tools to underperform with very high depth sequencing data? It might be of interest to comment on this in the discussion, since many studies dealing with clonal evolution modeling confirm presence of somatic mutations discovered using WES or WGS by ultra-high depth targeted sequencing.

Changes: We had another detailed look at the clustering results, comparing 1500x and 2000x. The discussion was extended, now also considering the aspect of very high coverage (the main reason seems to be estimated CCFs getting over-interpreted and thus leading to clusters being split).

Round 2

Reviewer 1 Report

Dear Sandamm and colleagues,

I would like to express my appreciation for responding to all of my comments in a valid and appropriate manner.

The revised version, in my opinion, is more focused, robust, and qualitatively superior.

In my opinion, the manuscript has met the requirements for publication in IJERPH.

Thank you.

Best regards